# Computational drug repurposing against SARS-CoV-2 reveals plasma membrane cholesterol depletion as key factor of antiviral drug activity

**Szilvia Barsi**[1], **Henrietta Papp**[2,3], **Alberto Valdeolivas**[4], **Dániel J. Tóth**[1], **Anett Kuczmog**[2,3], **Mónika Madai**[2,3], **László Hunyady**[1,5,6], **Péter Várnai**[1,5], **Julio Saez-Rodriguez**[4], **Ferenc Jakab**[2,3], **Bence Szalai**[1]¤*

**1** Semmelweis University, Faculty of Medicine, Department of Physiology, Budapest, Hungary, **2** National Laboratory of Virology, University of Pécs, Pécs, Hungary, **3** Institute of Biology, Faculty of Sciences, University of Pécs, Pécs, Hungary, **4** Heidelberg University, Faculty of Medicine, and Heidelberg University Hospital, Institute for Computational Biomedicine, Bioquant, Heidelberg, Germany, **5** MTA-SE Laboratory of Molecular Physiology, Budapest, Hungary, **6** Institute of Enzymology, Research Centre for Natural Sciences, Budapest, Hungary

¤ Current address: Turbine Simulated Cell Technologies Ltd., Budapest, Hungary
* ben.szalai@gmail.com

**Data Availability Statement:** All relevant data are within the manuscript and its Supporting Information files. All analysis code to reproduce the

## Abstract

Comparing SARS-CoV-2 infection-induced gene expression signatures to drug treatment-induced gene expression signatures is a promising bioinformatic tool to repurpose existing drugs against SARS-CoV-2. The general hypothesis of signature-based drug repurposing is that drugs with inverse similarity to a disease signature can reverse disease phenotype and thus be effective against it. However, in the case of viral infection diseases, like SARS-CoV-2, infected cells also activate adaptive, antiviral pathways, so that the relationship between effective drug and disease signature can be more ambiguous. To address this question, we analysed gene expression data from *in vitro* SARS-CoV-2 infected cell lines, and gene expression signatures of drugs showing anti-SARS-CoV-2 activity. Our extensive functional genomic analysis showed that both infection and treatment with *in vitro* effective drugs leads to activation of antiviral pathways like NFkB and JAK-STAT. Based on the similarity—and not inverse similarity—between drug and infection-induced gene expression signatures, we were able to predict the *in vitro* antiviral activity of drugs. We also identified SREBF1/2, key regulators of lipid metabolising enzymes, as the most activated transcription factors by several *in vitro* effective antiviral drugs. Using a fluorescently labeled cholesterol sensor, we showed that these drugs decrease the cholesterol levels of plasma-membrane. Supplementing drug-treated cells with cholesterol reversed the *in vitro* antiviral effect, suggesting the depleting plasma-membrane cholesterol plays a key role in virus inhibitory mechanism. Our results can help to more effectively repurpose approved drugs against SARS-CoV-2, and also highlights key mechanisms behind their antiviral effect.

results of this manuscript is available at https://github.com/comp-sys-pharm/SARS-CoV-2-cholesterol.

**Funding:** BS was supported by the Premium Postdoctoral Fellowship Program of the Hungarian Academy of Sciences (460044). DJT and PV were supported by the Hungarian Scientific Research Fund (OTKA K134357). On behalf of Project DRUGSENSPRED we thank for the usage of ELKH Cloud (https://science-cloud.hu/) that significantly helped us achieve the results published in this paper. The in vitro SARS-CoV-2 experiments were funded by the Hungarian Scientific Research Fund (OTKA KH129599), by the European Union and the European Social Fund (EFOP-3.6.1.-16-2016-00004), and by the Ministry for Innovation and Technology of Hungary (TUDFO/47138/2019-ITM) to FJ. Also, project no. TKP2021-NVA-07 has been implemented with the support provided from the National Research, Development and Innovation Fund of Hungary, financed under the TKP2021-NVA funding scheme to FJ. The funders had no role in study design, data collection and analysis, decision to publish, or preparation of the manuscript.

**Competing interests:** I have read the journal's policy and the authors of this manuscript have the following competing interests: JSR reports funding from GSK and Sanofi and fees from Travere Therapeutics and Astex. BS is a full time employee of Turbine Ltd., Budapest, Hungary.

## Author summary

Targeting the infected host cells is an effective strategy in infectious diseases, like COVID-19. Better understanding the virus and drug induced cellular mechanisms can help to identify new compounds with potential antiviral activity. We used computational methods to analyse gene expression data from *in vitro* SARS-CoV-2 infected cell lines, and gene expression signatures of drugs showing anti-SARS-CoV-2 activity. With the help of machine learning methods, we were able to predict *in vitro* effective antiviral drugs from gene expression based features. We found that effective drugs activate antiviral pathways like JAK-STAT and NFkB, and also the SREBF transcription factors, key regulators of cholesterol synthesis. Using microscopic measurements we validated that several antiviral drugs influence the cholesterol content of the plasma membrane. Finally, we showed that cholesterol rescue inhibited the *in vitro* antiviral effect of amiodarone, demonstrating the importance of drug induced cholesterol changes in the antiviral drug effect.

## 1. Introduction

The newly emerged Severe Acute Respiratory Syndrome Coronavirus 2 (SARS-CoV-2), causing the coronavirus disease 2019 (COVID-19), has led to more than 420,000,000 reported infections and 5,500,000 reported deaths worldwide [1] until February 2022. Identification of new therapeutic compounds against SARS-CoV-2 / COVID-19 is an urgent need until effective vaccination is worldwide available and given the emergence of SARS-CoV-2 strains showing immune evasion [2]. The main therapeutic strategies include A) inhibiting key viral enzymes (like remdesivir [3]); B) modulating the infected cells to decrease viral replication [4,5] and C) modulating the over-activation of the immune system to treat late complications like "cytokine storm" [6–8]. Repurposing already approved drugs for these indications is especially important as it allows a shorter time of approval for anti-SARS-CoV-2 treatment.

Comparing gene expression signatures of drugs and diseases have been previously shown to be an effective strategy to repurpose drugs for new therapeutic indications [9]. The general principle of these studies is that a drug inducing an opposite gene expression signature to a disease signature can reverse the disease-related gene expression changes, thus the disease phenotype. This "signature reversal" principle has also been used to predict effective drugs against SARS-CoV-2 infection [10–12]. However, these predictions lack, in most cases, mechanistic insight and experimental validation. Moreover, as infected cells activate adaptive antiviral pathways (like interferon pathway), inhibiting these pathways does not necessarily decrease viral replication.

In this study, we analyzed transcriptomics data from *in vitro* SARS-CoV-2 infected cell lines (section 2.1) and from cell lines treated with drugs showing anti-SARS-CoV-2 activity (effective drugs, section 2.2). Functional genomic analysis revealed shared transcription factor and pathway activity changes (eg. increased activity NFkB and JAK-STAT pathways) in the infected and effective drug-treated cell lines. Similarity between infection signature and drug-induced signature was predictive for *in vitro* effective drugs, contradictory to the classical "signature reversal" principle (section 2.3). Machine learning-based prediction of effective drugs identified SREBF1 and SREBF2 transcription factors, key regulators of lipid metabolism, as important factors of antiviral drug effect. Using a fluorescently labeled cholesterol sensor, we showed the decreased level of plasma-membrane cholesterol in cells treated with effective drugs, like chlorpromazine, confirming the effect of these drugs on cholesterol metabolism (section 2.4). We also identified amiodarone, a drug decreasing plasma-membrane cholesterol

content, thus a potential *in vitro* effective drug. Using an *in vitro* SARS-CoV-2 infection assay, we demonstrated that the antiviral effect of amiodarone can be reversed by cholesterol supplement, underlying the relevance of decreased plasma-membrane cholesterol in the antiviral drug effect (section 2.5).

## 2. Results

### 2.1 Analysis of host pathway and transcription factor activities reveals adaptive response of SARS-CoV-2 infected cells

We analysed gene expression data from two recent studies (GSE147507 [13] and GSE148729 [14]), where lung epithelial cancer cell lines (Calu-3 and A549) were infected with SARS-CoV-2. To identify infection-induced pathway and transcription factor (TF) changes, we used the PROGENy [15,16] and DoRothEA [17,18] tools, respectively (more details in Methods).

PROGENy analysis showed increased activity of NFkB and TNFa pathways in both analysed cell lines, while the activity of JAK-STAT pathway increased more pronounced in infected Calu-3 cell lines (Fig 1A). DoRothEA analysis (Fig 1B) revealed strong activation of STAT, IRF and NFkB transcription factors, while cell growth-related transcription factors (E2Fs, Myc) showed decreased activity. Also SREBF1/2, key transcriptional regulators of cholesterol synthesis, showed decreased activity. STATs, IRFs and NFkB pathways / TFs play a key role in antiviral innate immunity [19]. Decreased activity of E2Fs and Myc [20] and decreased synthesis of cholesterol [21] are also part of the physiological antiviral / interferon response.

To further analyse which upstream signalling pathways regulate the inferred TF activity changes, we used CARNIVAL [22], a signaling network contextualisation tool, which connects transcription factor activities to perturbations in signaling networks via integer linear programming (more details in Methods). We performed CARNIVAL analysis using inferred transcription factor activities from a SARS-CoV-2 infected cell line (GSE147507, Calu-3), and used RIG-I like receptors (DDX58 and IFIH1), key receptors for foreign RNA sensing [23], as main perturbation target. CARNIVAL results showed (Fig 1C), that activation of RIG-I like receptors by the dsRNA of SARS-CoV-2 can directly lead to the observed transcription factor activity changes, including activation of NFkB, IRFs and STATs and inhibition of SREBF2 and E2F4. Key identified intermediate nodes AKT1 and MAPK1 were already connected to coronavirus infection [5,24] and other viral infections [25,26], also suggesting that the observed TF changes are initiated by the RIG-I like receptors, thus corresponding to the antiviral response of the host cell.

In summary, our functional analysis of the gene expression changes in SARS-CoV-2 infected cell lines suggests that a large part of the induced pathway / transcription factor activity changes are adaptive, i.e. part of the physiological antiviral response.

### 2.2 Analysis of *in vitro* anti-SARS-CoV-2 drug-induced pathway and transcription factor activities reveals similar changes to virus infection

To compare infection and drug-induced signatures, we used a large compendium of drug-induced gene expression signatures from the LINCS-L1000 project [27]. LINCS-L1000 contains drug-induced gene expression signatures from different cell lines, concentrations and time points. We calculated consensus gene signatures for each drug using our previous approach ([28], Methods), ending up with gene expression signatures for 4671 drugs. To select drugs effectively inhibiting SARS-CoV-2 replication *in vitro*, we used a curated database created by ChEMBL (http://chembl.blogspot.com/2020/05/chembl27-sars-cov-2-release.html). This dataset contains 133 drugs previously showing effective inhibition of viral replication in 8

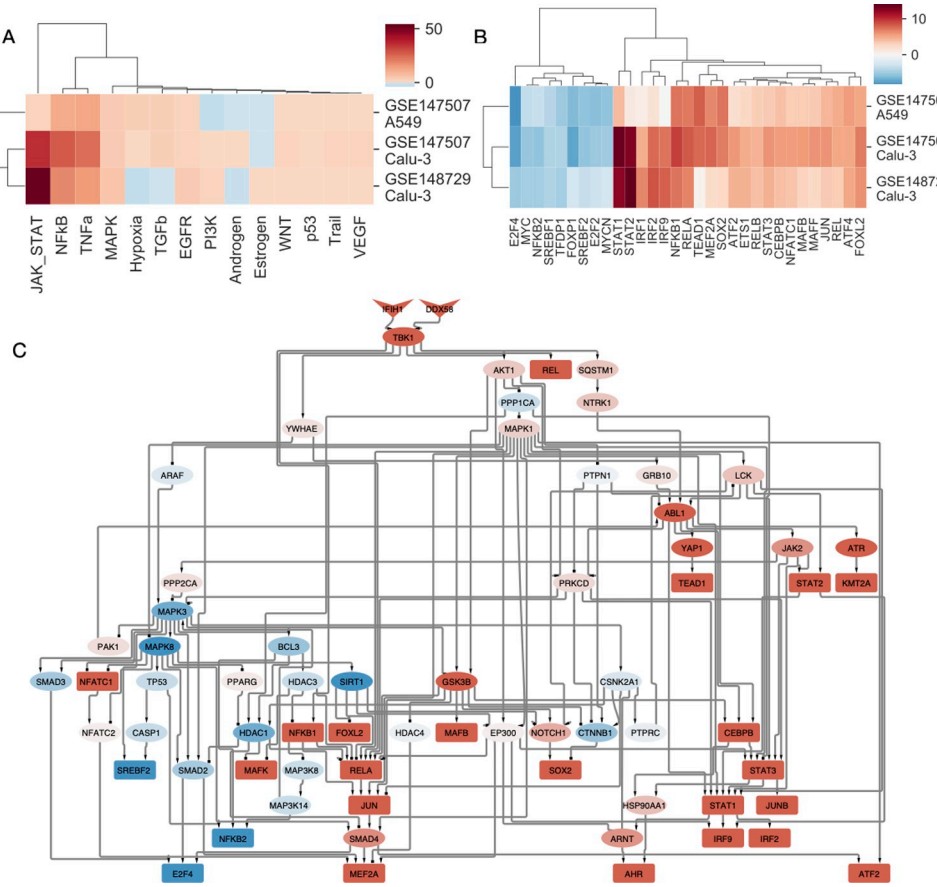

**Fig 1. Functional genomic analysis of SARS-CoV-2 infected cell lines.** (A) Inferred pathway and (B) TF activities of SARS-CoV-2 infected samples from lung epithelial cell lines (Calu-3 and A549). Activities were calculated from differential expression signatures (infected—control) using PROGENy and DoRothEA tools for pathway and TF activities, respectively. Only TFs with high absolute level of activity changes (absolute normalized enrichment score > 4) are shown. (C) Causal signalling network in SARS-CoV-2 infected Calu-3 cells (GSE147507) identified by CARNIVAL. RIG-I like receptors (DDX58 and IFIH1) as perturbation targets and DoRothEA inferred TF activities were used as the input of the CARNIVAL pipeline. Color code represents inferred activity of protein nodes (blue: inhibited, red: activated).

studies [4,29–35]. We found an intersection of 47 drugs between LINCS-L1000 (available gene expression signatures) and ChEMBL dataset (*in vitro* effective drugs). To characterize drug-induced pathway and transcription activity changes, we analysed consensus drug-induced signatures using PROGENy and DoRothEA.

PROGENy analysis showed strong activation of NFkB and TNFa pathways by several drugs, including niclosamide, perhexiline and digoxin (Fig 2A). Several drugs also strongly activated the JAK-STAT pathway (RTK inhibitors osimertinib and regorafenib). In case of TF analysis, we found similar patterns (Fig 2B) to the infection-induced signatures: increased activity of NFkB and STAT transcription factors and decreased activity of Myc/E2Fs transcription factors. Interestingly, SREBF1/2 showed strongly increased activity for a large cluster of drugs, but (similar to the infection signatures) decreased in another cluster. To further analyse the TF activity changes in the different clusters of drugs, we calculated average TF activities for these clusters and plotted these values against the average TF activities of the 3 SARS-CoV-2 infection signatures (Fig 2C). One cluster (Fig 2C, upper left panel), showed high correlation (Spearman's rho = 0.64, p = 8.55e-35) across all TFs. Two other clusters (Fig 2C, upper middle

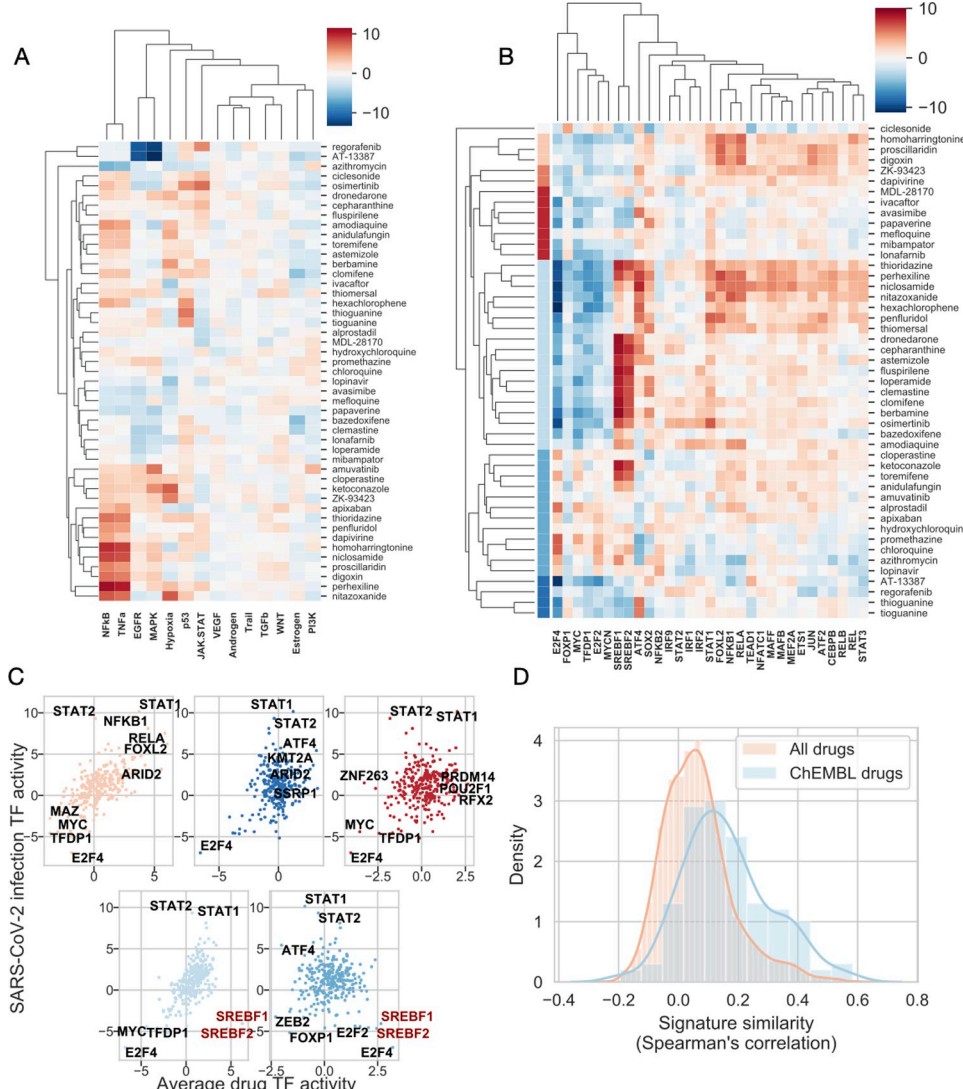

**Fig 2. Functional genomic analysis of effective drugs treated cell lines.** (A) Inferred pathway and (B) TF activities of anti-SARS-CoV-2 drug-treated cell lines. Activities were calculated from LINCS-L1000 consensus drug-induced signatures, using PROGENy and DoRothEA tools for pathway and TF activities, respectively. Drug clusters in (B) are color coded. Only selected transcription factors (corresponding to Fig 1B) are shown. (C) Relationship between average TF activities induced by drug treatment and SARS-CoV-2 infection for 5 different drug clusters (colors of clusters correspond to panel B). TFs with the highest/lowest average activities are text labeled. (D) Density plot of similarities between SARS-CoV-2- and drug-induced signatures for all LINCS-L1000 drugs and known anti-SARS-CoV-2 drugs (ChEMBL drugs).

and upper right panels) showed lower, but still significant correlation with infection TF activity signature (Spearman's rho = 0.14 and 0.18, p = 0.0122 and 0.00174, respectively), with prominent increase of STATs and decrease of E2F4 transcription factor activity. For the remaining two large clusters, we found either negligible (Fig 2C, lower right panel) or high (Fig 2C, lower left panel) correlation with infection-induced TF activities (Spearman rho = 0.04 and 0.58, p = 0.484 and 3.14e-27, respectively), but we found high drug-induced activity of SREBF1/2 transcription factors in these clusters, opposite to the inhibition of these TFs by SARS-CoV-2 infection.

As we found that, for several drug clusters, drug-induced TF activities showed positive correlation with SARS-CoV-2-induced TF activities, we were interested in the general similarity of drug and infection-induced gene expression signatures. To achieve this we calculated the signature similarity (Spearman's correlation coefficient, which has been previously shown to be an effective metric to analyse signature similarity for the LINCS-L1000 data [27,28]) between all the 4,671 drug-induced signatures from our LINCS-L1000 dataset and the infection signatures. We found that effective anti-SARS-CoV-2 drugs (ChEMBL dataset) have higher similarity to infection signatures, than ineffective drugs / drugs with unknown efficacy (Fig 2E, Mann-Whitney U test p-value = <1e-200).

In summary, we found that known *in vitro* effective anti-SARS-CoV-2 drugs induce similar pathway and TF activity patterns, and appropriately similar gene expression signatures to virus infection signatures. We also identified two large clusters of drugs inducing strong activation of SREBF1/2 transcription factors, key regulators of cholesterol / lipid metabolism.

## 2.3 Prediction of drugs with *in vitro* anti-SARS-CoV-2 activity

After identifying some general patterns in the gene expression signatures of *in vitro* effective anti-SARS-CoV-2 drugs, we investigated how well we can predict drug effectiveness using gene expression signatures.

As a first strategy, we simply used the previously calculated drug—infection signature similarity to predict effective drugs. Using these similarity values (predicted score) and the known *in vitro* effective drugs (ChEMBL dataset, true positive values) we performed ROC analysis (Fig 3A). We found that similarity to infection signatures is predictive for effective drugs, i.e. drugs with high similarity to infection signature are more frequently effective (ROC AUCs: 0.75, 0.74 and 0.64 for GSE147507 A549, GSE147507 Calu-3 and GSE148729 Calu-3, respectively). To test the specificity of this signature similarity-based approach for SARS-CoV-2 infection signature, we included several other virus infection-induced gene expression signatures for SARS-CoV (GSE33267 [36], GSE148729), MERS (GSE45042 [37], GSE56677 [38]), respiratory syncytial virus (RSV, GSE147507), influenza (GSE28166 [39], GSE37571) and human parainfluenza (HPIV, GSE147507) infected Calu-3 and/or A549 cell lines. Similarity to these infection signatures showed lower predictive performance for anti-SARS-CoV-2 drugs (ROC AUC values <0.7 except one SARS and RSV signature with ROC AUCs 0.70 and 0.71, respectively, Fig 3B), suggesting the relative SARS-CoV-2 specificity of the similarity-based methods.

Following this unsupervised prediction strategy, we also performed supervised, machine learning-based predictions. We used the drug-induced TF activities as features, and effective drugs from the ChEMBL dataset as positive examples, with Random Forest Classification as prediction algorithm. We set up a random subsampling based cross-validation scheme and evaluated the performance using ROC analysis (Methods). Our results showed a slightly improved performance compared to the unsupervised, similarity-based approach (mean ROC AUCs: 0.72 and 0.68, 0.66, 0.57, respectively for the machine learning and similarity-based methods, paired t-test p-values between machine learning and similarity-based methods: 3.02e-07, 2.76e-15, 4.89e-15 for GSE147507 A549, Calu-3 and GSE148729 Calu-3 signatures respectively, Fig 3C). To gain some more mechanistic insight from the prediction of machine learning models, we analysed feature importances (Gini importance, Fig 3D) of the Random Forest Regression models and found that SREBF1 and SREBF2 activity were the two most important features, followed by TFAP2A, HNF4A and TP63 transcription factors.

In summary, our two different prediction approaches showed reasonable performance (comparable to studies based on network medicine and chemical similarity [40,41]), to predict

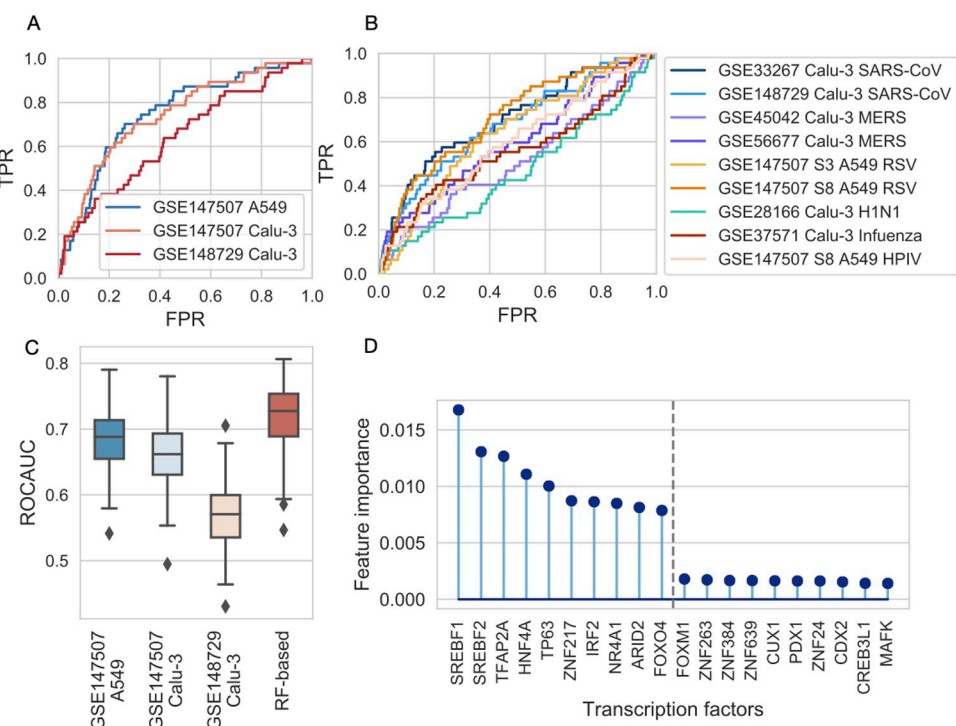

**Fig 3. Evaluation of similarity-based and machine learning-based models in predicting *in vitro* effective drugs.** (A, B) ROC analysis of similarity-based predictions of effective drugs against SARS-CoV-2. Drug—SARS-CoV-2 (A) or drug—other virus (B) infection signature similarity was used as prediction score, while known *in vitro* effective drugs (ChEMBL dataset) were used as true positives. (FPR: false positive rate, TPR: true positive rate) (C) Comparison of predictive performance (ROCAUCs) of similarity-based method (similarity to SARS-CoV-2 infection signature, x-axis) and random forest-based (RF-based, x-axis) prediction. Results of 100 random subsampling cross-validations. In case of similarity-based methods, ROC AUC curves were only calculated for the corresponding cross-validation sets. Boxplots represent the median (central line), first and third quartile (box), minimum and maximum non-outlier values (whiskers) and outliers (diamonds). (D) Feature importances (Gini importance) of the Random Forest model. Top and bottom 10 features (TFs) are shown according to importance.

drugs with *in vitro* anti-SARS-CoV-2 activity, and also highlighted the importance of previously discussed SREBF1/2 transcription factors. Drug—SARS-CoV-2 signature similarities, and predicted probabilities of anti-SARS-CoV-2 activity is available in S1 Table.

## 2.4 Anti-SARS-CoV-2 drugs are increasing SREBF activity by depleting plasma membrane cholesterol

While in most cases we found similarity between the activity of SARS-CoV-2 infection and *in vitro* effective drug-induced transcription factor activities, in case of SREBF1/2 we found opposite changes: SARS-CoV-2 infection inhibited SREBF1/2, while a large cluster of effective drugs lead to increased activity of SREBFs. SREBFs are activated through the decreased cholesterol content of plasma membrane and endoplasmic reticulum, and activated SREBFs induce the expression of cholesterol, and other lipid synthesizing enzymes [42]. From this point of view, decreased SREBF activity during viral infection can lead to decreased cholesterol synthesis, which can inhibit the viral replication and/or viral entry [21], thus can be considered as an adaptive response of the host cell (Fig 4A). Interestingly, we observed a strongly increased SREBF activity in large clusters of effective drugs. To resolve this discordance, we hypothesized that these *in vitro* effective drugs directly decrease plasma membrane cholesterol (Fig 4A). In this case, drug-induced decrease of plasma membrane cholesterol can contribute to the

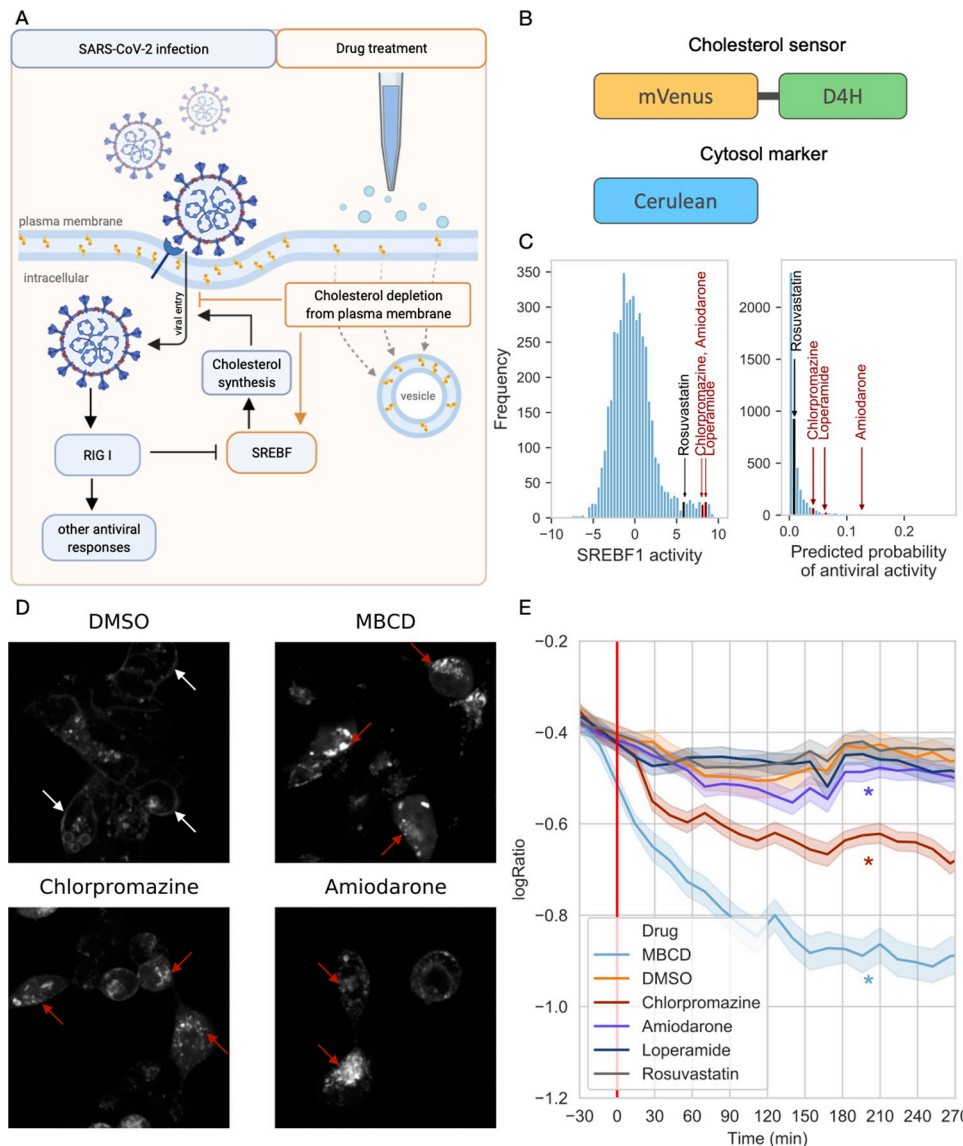

**Fig 4. Cholesterol depleting effect of SREBF activating drugs.** (A) Schematic figure of the hypothesis that antiviral drugs block virus entry into cells by cholesterol depletion from plasma membrane, and are leading to a compensatory increased SREBF1/2 activity. Effects induced by viral infection are marked with black arrows (left side), while orange arrows represent drug-induced changes (right side) The figure was created with BioRender.com. (B) Schematic representation of the used fluorescent constructs. (C) Histogram of SREBF1 activation (left panel) and histogram of predicted probabilities of *in vitro* antiviral activity of LINCS-L1000 drugs (right panel, according to the Random Forest model). Drugs selected for *in vitro* experiments are text labeled. (D) Representative confocal microscopy images of D4H-mVenus transfected HEK293A cells treated with DMSO, MβCD, chlorpromazine or amiodarone. White arrows mark plasma membrane, while red arrows show intracellular localised cholesterol sensors. (E) Time-dependent change of $\log_2$(PM/IC) ratio of average cholesterol sensor intensity in HEK293A cells treated with DMSO, MβCD, chlorpromazine, amiodarone, loperamide or rosuvastatin. Red line marks drug treatment. *: significant (p<0.001) interaction between drug treatment and elapsed time in linear model.

antiviral effect, while decreased cholesterol levels can activate SREBFs, thus explaining the observed increased activity of these TFs in our bioinformatic analysis.

To confirm this hypothesis, we performed high-throughput, automatic confocal micros-copy imaging using a fluorescently labeled cholesterol sensor domain, D4H-mVenus [43,44].

HEK293A cells were co-transfected with D4H-mVenus and cytoplasmic Cerulean as cytosolic marker (Fig 4B), and treated with dimethyl sulfoxide (DMSO, negative control), MβCD (methyl β-cyclodextrin, plasma membrane cholesterol depleting compound, as positive control) and 3 drugs from our computational drug repurposing pipeline, loperamide, amiodarone and chlorpromazine (all drugs were used in 10 μM final concentration). All these three drugs increased the activity of SREBF transcription factors (Fig 4C, left panel). Loperamide and chlorpromazine have been previously shown to be *in vitro* effective against SARS-CoV-2 (ChEMBL dataset), while amiodarone was one of the top predicted drugs of the Random Forest model (Fig 4C, right panel, ranked 36/4671 drugs, S1 Table). We also treated HEK293A cells with rosuvastatin, an inhibitor of cholesterol synthesis. Rosuvastatin also alters cellular cholesterol metabolism, however, it does not influence plasma membrane cholesterol directly, but inhibits HMG-CoA reductase, the rate limiting enzyme of *de novo* cholesterol synthesis. Rosuvastatin was not predicted as an effective anti-SARS-CoV-2 drug by the Random Forest model (Fig 4C left panel, ranked 1821/4671 drugs).

Cells were treated with the different drugs and serial confocal microscopy images were recorded for 4.5 hours. In untreated, or DMSO treated cells, we observed a predominantly plasma membrane localisation of the fluorescent protein labeled cholesterol sensor (Fig 4D, top left panel). Treatment with MβCD led to decreased plasma membrane cholesterol levels, while cholesterol accumulated in intracellular vesicles (Fig 4D, top right panel). We observed similar phenotypic changes in case of amiodarone and chlorpromazine (Fig 4D, bottom panels), while the localisation of cholesterol sensor in loperamide and rosuvastatin treated cells was more similar to control condition (S1 Fig).

For a more systematic and unbiased analysis of the changes in the localisation of cholesterol sensors, we performed quantitative image analysis (S2 Fig). For each cell in each image, we calculated the ratio of average plasma membrane (PM) and average intracellular (IC) D4H-mVenus fluorescence (PM/IC ratio). To segment cells in confocal microscopy images, we used *Cellpose* library ([45], Methods). Plotting the PM/IC ratio as a function of elapsed time after drug treatment (Fig 4E) revealed that PM/IC ratio did not decrease in loperamide and rosuvastatin treated samples, while MβCD, chlorpromazine and amiodarone treatment induced significant decrease of the ratio (linear model coefficients values for interaction between drug treatment and time: -0.002, -0.00086, -0.00017, -0.000032 and 0.000083 for MβCD, chlorpromazine, amiodarone, loperamide, rosuvastatin respectively, p values: <1e-200, <1e-200, 2.71e-09, 0.25 and 0.0047), confirming the plasma membrane cholesterol depleting effect of chlorpromazine and amiodarone, two SREBF activating drugs.

In summary, our high-throughput image acquisition and analysis pipeline confirmed that chlorpromazine and amiodarone decreased plasma membrane cholesterol content, which explains the increased activity of SREBF transcription factors in case of gene expression readout.

## 2.5 Supplementing cholesterol reverses anti-SARS-CoV-2 activity of amiodarone

As our experiments revealed that the selected drugs with *in vitro* anti-SARS-CoV-2 activity decreased the cholesterol content of plasma membrane, we were interested in whether decreased plasma membrane cholesterol levels could play a causal role in the antiviral effect, according to our assumptions (Fig 4A). To test this hypothesis, we performed *in vitro* SARS-CoV-2 viral infection assay with cholesterol rescue in Vero-E6 cells.

At first we tested whether the investigated drugs show anti-SARS-CoV-2 activity in our previously described experimental system [46]. Briefly, Vero-E6 cells were co-treated with

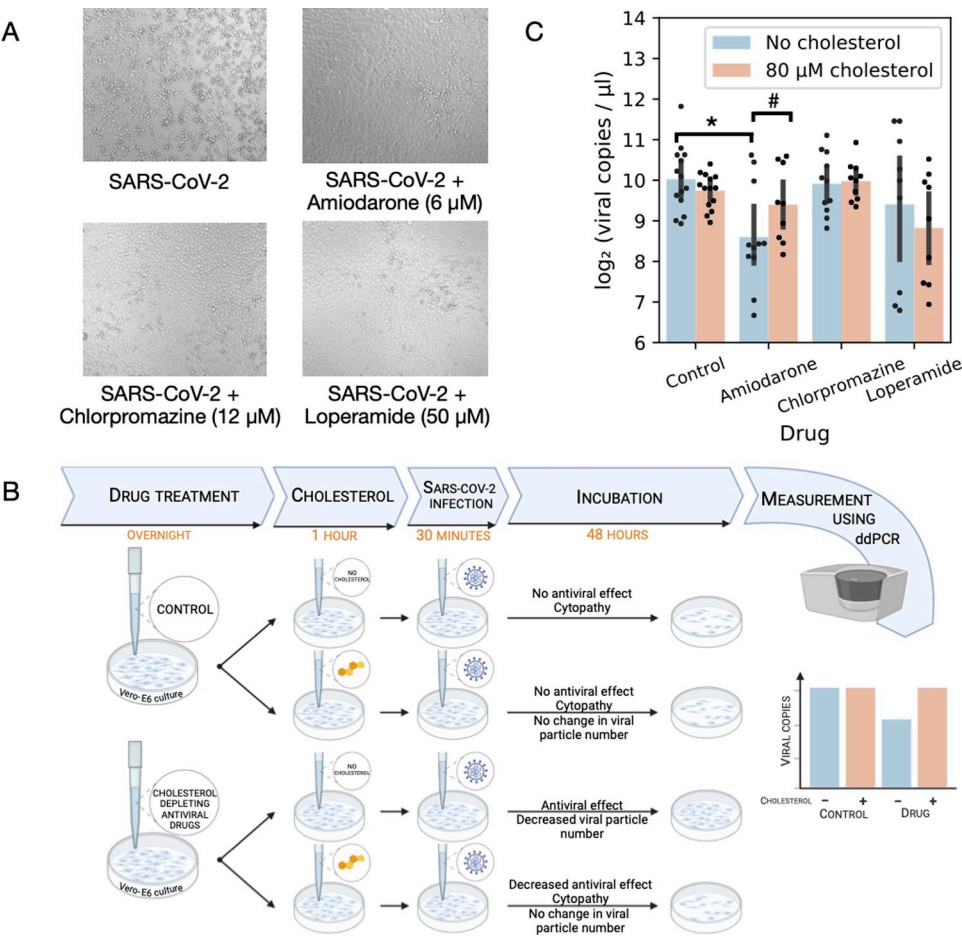

**Fig 5. Cholesterol replenishment inhibits antiviral effect of amiodarone.** (A) Predicted drugs inhibit SARS-CoV-2 replication in infected Vero-E6 cells. Vero-E6 cells were infected with SARS-CoV-2 (top left) and co-treated either with amiodarone (top right), chlorpromazine (bottom left) or loperamide (bottom right). Antiviral effect (reduced cytopathy) was evaluated by microscopic imaging (10x objective) 48 hours after infection. (B) Schematic figure of cholesterol rescue experiments. The figure was created with BioRender.com. (C) Effect of cholesterol rescue on antiviral drug effect. Vero-E6 cells were pretreated with drugs (x-axis), cholesterol was replenished (color code) and cells were infected with SARS-CoV-2. Antiviral effect of drugs was evaluated 48 hours after infection by droplet digital PCR (viral copies, y-axis). *, #: significant (p<0.05) effect of drug treatment and drug treatment-cholesterol interaction in linear model, respectively.

SARS-CoV-2 and the selected drugs (6 μM amiodarone, 12 μM chlorpromazine or 50 μM loperamide, effective drug concentrations were selected based on preliminary experiments) for 30 minutes, then washed and incubated with the drugs for 48 hours. Infection efficacy was evaluated by microscopic examination of infection-induced cytopathic effect (CPE, more details in Methods). Untreated, SARS-CoV-2 infected cells showed strong cytopathy (Fig 5A, top left panel), while amiodarone, chlorpromazine and loperamide markedly reduced the infection-induced cytopathy, confirming the antiviral effect of these drugs (Fig 5A). The used compounds did not lead to cellular toxicity in the used concentrations (S3 Fig).

To test the effect of plasma membrane cholesterol depletion on SARS-CoV-2 infectivity, we performed cholesterol rescue experiments (Fig 5B). Vero-E6 cells were treated with drugs overnight, then the media was replaced with cholesterol (80 μM) containing media. After 1 hour of cholesterol treatment, the cells were infected for 30 min with SARS-CoV-2. Infection efficacy was evaluated 48 hours after infection by droplet digital PCR based viral RNA

quantification (Fig 5C). Chlorpromazine and loperamide did not have antiviral effect in the pretreatment setting (linear model p values: 0.62 and 0.18, respectively), while amiodarone decreased viral particle number significantly (linear model p-value: 1.47e-05). Cholesterol replenishment significantly increased viral particle number in amiodarone treated Vero-E6 cells (amiodarone: cholesterol interaction term p-value: 0.026), confirming the causal role of drug-induced cholesterol depletion in the antiviral effect of amiodarone.

Our *in vitro* SARS-CoV-2 infection assay confirmed the antiviral effects of chlorpromazine, loperamide and amiodarone, and cholesterol rescue experiments suggest that plasma membrane cholesterol depletion plays an important role in the antiviral effect of amiodarone.

## 3. Discussion

In this study, we analysed the gene expression signatures of *in vitro* SARS-CoV-2 infected cells and effective anti-SARS-CoV-2 drugs. Using functional genomic computational tools, we showed that both virus infection and drug treatment leads to similar changes of pathway and transcription factor activities, like activation of antiviral NFkB and JAK-STAT pathways. Signature similarity between infection and drug-induced signature was predictive for drugs with *in vitro* anti-SARS-CoV-2 activity, contrary to the classical "signature reversal" hypothesis. Using machine learning models we effectively predicted anti-SARS-CoV-2 drugs, and predicted amiodarone as an *in vitro* antiviral compound. More detailed functional genomic analysis of TF activities revealed that SREBF1/2 TFs are strongly activated by large clusters of effective drugs. Using a high-throughput confocal microscopy setup and quantitative image analysis we showed that two of the three investigated effective drugs influence cellular distribution of cholesterol, leading to decreased plasma membrane cholesterol content. Viral infection assay confirmed the already described *in vitro* antiviral activity of loperamide and chlorpromazine, and also the predicted antiviral activity of amiodarone. Cholesterol supplement reversed the antiviral effect of amiodarone, suggesting the causal role of decreased membrane cholesterol in the antiviral effect.

Gene expression-based computational drug repurposing is a promising field to find new disease indications of existing drugs [47]. Despite its simplicity, it has been used successfully to identify repurposable drugs for different diseases from cancer [48,49] through inflammatory [50] to metabolic [51] diseases. While most of the related works rely on the "signature reversal" hypothesis, in case of infection diseases, like COVID-19, it is less clear whether signature reversal (inhibiting the virus-hijacked signalisation) or signature similarity (promoting the antiviral response of infected host cells) can be more effective. While early studies at the beginning of the COVID-19 pandemic applied mostly the original signature reversal hypothesis, more recent works [52,53] also assumed that drugs with similarity to the SARS-CoV-2-induced gene expression signature can be effective. In our work, we performed a more unbiased analysis of signature-based drug repurposing against SARS-CoV-2. We compared the gene expression signatures of known effective drugs against SARS-CoV-2 infection signatures, and found that signature similarity, and not dissimilarity, is predictive for antiviral effect. These results suggest that increasing the antiviral response of host cells can be a more effective strategy than inhibiting viral infection-induced pathways. Whether this is specific for SARS-CoV-2, known for evading several antiviral systems of the host cell [54], or a general mechanism for (viral) infections, needs further analysis with large scale *in vitro* drug screenings against other viruses. Nevertheless, using our "signature similarity" principle instead of—or together with—the "signature reversal" hypothesis can accelerate computational drug repurposing against existing and emerging infectious diseases, complemented by network-based repurposing strategies [40,55].

While signature (dis)similarity-based computational drug repurposing has promising predictive performance, it gives no real mechanistic insight. To overcome this problem, we performed extensive functional genomic analysis of SARS-CoV-2 and drug-induced gene expression signatures. We found that both viral infection and effective drugs stimulate known antiviral pathways like NFkB and JAK-STAT. We observed lower induction of these pathways in virus-infected A549 cells, compared to Calu-3 cell lines, probably based on the lower expression of virus receptor ACE2 in A549 cells. The activation of antiviral pathways in virus-infected and effective drug-treated cells also supports the "signature similarity" principle.

Beside the activation of antiviral TFs and pathways, we also observed inhibition of (inferred) SREBF1/2 transcription factors in SARS-CoV-2 infected samples, while an activation of these TFs in a large cluster of antiviral drug-treated cells. SREBF1/2 regulate the expression of key members of cholesterol synthesis. Cholesterol depletion of plasma membrane can reduce SARS-CoV-2 infection [56,57], and decreasing SREBFs activity (and cholesterol synthesis) can be also part of the physiological, interferon-induced antiviral response of the host cell [21,58,59]. Recently, inhibitors of the SREBFs—DNA interaction were found to exert antiviral effects [60] and CRISPR based knock-out of the SREBF pathway members also led to SARS-CoV-2 resistant phenotype [61], suggesting that inhibition of SREBFs could be beneficial in case of SARS-CoV-2 infection. In contrast, we found increased activity of SREBFs in case of several effective drug-induced gene expression signatures. Previous works also showed the increased expression of lipid metabolic enzymes [12] in antiviral drug-treated cells, and a recent large scale CRISPR screen [62] also found that increased cholesterol synthesis can reduce SARS-CoV-2 infection. However, these two later conclusions were based on the analysis of gene expression changes of the cholesterol synthetic pathway. Gene expression changes are in several cases not the cause, but the (compensatory) consequence of perturbed cell states [63,64]. Based on this, we hypothesized that increased SREBF1/2 activity (based on transcriptional readout) can be a compensatory consequence of decreased plasma membrane cholesterol levels in case of several antiviral drugs. Using a fluorescent cholesterol sensor, we found that amiodarone and chlorpromazine, two effective *in vitro* antiviral drugs, indeed decreases the cholesterol content of plasma membranes, which can explain the (compensatory) increased SREBF1/2 activity. In an *in vitro* SARS-CoV-2 infection assay, coupled with cholesterol rescue, we also showed that cholesterol replenishment reduced the antiviral activity of amiodarone, thus confirmed the causal role of plasma membrane cholesterol decrease in the antiviral effect of amiodarone. While our computational analysis also predicted that PM cholesterol depletion plays a role in the antiviral effect of chlorpromazine and loperamide, we were not able to verify these predictions experimentally. Noteworthy, these two drugs had antiviral effect in case of co-treatment with virus infection, but not in the case of the pre-treatment setup used in cholesterol rescue experiments (probably due to pharmacokinetic factors). It is thus hard to draw conclusions about the role of cholesterol in the antiviral effect of these drugs.

While we showed that PM cholesterol depletion can be an important factor in the *in vitro* antiviral effect of drugs, whether this can be translated to *in vivo* is still an open question. A recent large scale study [65] showed that several *in vitro* repurposable drugs exert their antiviral effect via altering the membrane composition of drug-treated cells, and this antiviral effect has low translation potential based on concerns regarding drug concentration and adverse effects. While the authors of this study concluded that *phospholipidosis* is the main drug-induced membrane component change, our results argue that altered cholesterol content can also be a causal factor in the antiviral effect of drugs. Whether altered lipid composition of cellular membranes is only a factor confounding drug repurposing studies, or this effect can be exploited towards effective therapy, needs further studies.

In summary, our study showed that *in vitro* SARS-CoV-2 infection and effective antiviral drugs lead to similar pathway and transcription activity changes. We found that gene expression signature similarity, and not the dissimilarity, predicts *in vitro* effective antiviral compounds, which can accelerate computational drug repurposing against infectious diseases, and we made the results of our predictions available for the research community (S1 Table). We also identified that plasma membrane cholesterol depletion plays an important role in the mechanism of action of several antiviral drugs, and that cholesterol replenishment inhibits the *in vitro* antiviral effect of amiodarone, thus our results also give mechanistic insight about the antiviral effect of repurposable drugs.

## 4. Methods

### Virus infection-induced gene expression signatures

Microarray gene expression profiles of different virus-infected cell lines were downloaded from Gene Expression Omnibus (GEO) with accession numbers GSE28166 (H5N1), GSE37571 (Influenza), GSE33267 (SARS-CoV-1), GSE56677 and GSE45042 (MERS-CoV). Preprocessing and differential expression (DE) analysis was performed by using R package *limma* [66].

Total RNA-Seq profiles of SARS-CoV-2 and other virus-infected human cell lines were downloaded from GEO with accession numbers GSE147507 (SARS-CoV-2, RSV, IAV, HPIV) and GSE148729 (SARS-CoV-1 and 2). Differential expression (DE) analysis was performed using R library *DESeq2* [67].

In all gene expression datasets, we used (virus-infected—control) contrasts for differential expression calculation, where the control condition was mock infection. Where gene expression data after multiple time points were available, we used 24 h post-infection data. Shared genes across all datasets were selected and further analyzed.

### Drug treatment-induced signatures

We used Level 5 gene expression profiles from the LINCS-L1000 dataset [27]. We calculated consensus expression signatures for each drug (across different cell lines, concentrations and time points) using the MODZ method [27,28]. We matched LINCS-L1000 drugs with ChEMBL effective drug dataset (http://chembl.blogspot.com/2020/05/chembl27-sars-cov-2-release.html) using drug names and simplified molecular-input line-entry system (SMILES). Only measured (landmark) genes were used in the further analysis.

### Functional genomic analysis

From previously calculated SARS-CoV-2 infection and effective drug-induced signatures, we inferred pathway activities using PROGENy (R package *progeny* [15,16]) and transcription factor activities using DoRothEA (R package *dorothea* [18]).

PROGENy was applied to infer activities of 14 different pathways from expression and weight of their footprint gene sets. Z-scores of pathway activities were calculated using 10000 permutations of genes as background distribution. DoRothEA was applied to infer transcription factor activities using the *viper* algorithm [68]. DoRothEA is a collected, curated resource of signed TF-target interactions. Interactions are assigned a confidence level ranging from A (highest) to E (lowest) based on the number of supporting evidence. In this study interactions assigned A, B, C confidence levels were used. In transcription factor activity heatmaps (Figs 1B and 2B), only selected transcription factors (absolute normalised enrichment score > 4 in SARS-CoV–2 infection signatures) are shown.

We used the CARNIVAL tool [22] to contextualize our transcriptomics-based results into a mechanistic causal network. Briefly, CARNIVAL takes as input a prior knowledge network and a set of constraints and infers the most likely causal interactions by solving an integer linear programming problem. We assembled a curated prior knowledge signaling network from OmniPath resources [69]. As constraints, we selected the RIG-I like receptors (DDX58 and IFIH1) as upstream signaling perturbation and the top 25 most deregulated TFs (according to DoRothEA and *viper* results) upon SARS-CoV-2 infection as their downstream target. In addition, we used PROGENy pathway activity scores to weight the prior knowledge network and assist CARNIVAL in the discovery of optimal networks connecting the upstream perturbation (RIG-I like receptors) to the downstream targets (TFs).

## Signature similarity and machine learning-based prediction

We calculated similarities using Spearman's correlation between each virus infection-induced and each drug treatment-induced signature after selecting shared genes.

TF activity scores from drug-treated cells were used to predict effective drugs against SARS-CoV-2 using Random Forest Classifier from scikit-learn Python library [70]. The model was trained using 300 trees, with default parameters otherwise and with 100 different training sets. Training sets consisted of a 50% random sampling of effective drugs and non-effective drugs as well. The average importance of features (TFs) was computed (sum of feature importances, divided by the number of models). Predicted probabilities of antiviral activity were also computed in each prediction and the mean of them was calculated for each drug (probabilities were summed for each drug and divided by the number of occurrences in validation sets).

We performed ROC analysis using scikit-learn Python library to evaluate similarity-based and machine learning-based predictions. Effective drugs against SARS-CoV-2 curated by ChEMBL and overlapping with drugs of the LINCS-L1000 dataset were used as the positive class. The negative class consisted of the part of drugs from the LINCS-L1000 dataset not considered as effective by ChEMBL. To compare machine learning-based and similarity-based methods ROC curves were computed for each different validation set (100) and signature similarity scores of the corresponding drugs were considered.

## Fluorescent cholesterol sensor experiments

The cellular cholesterol sensor used in this study was the D4H domain [43,44] fluorescently labeled with monomer Venus (mVenus) on its N-terminus. To create the construct coding this sensor, we used a plasmid coding the bioluminescent version of the sensor (described in [71]), a kind gift from Tamas Balla (NICHD, NIH, Bethesda, USA). The D4H domain-coding sequence from this plasmid was subcloned into the pEYFP-C1 plasmid containing mVenus in place of EYFP, using BglII and BamHI restriction enzymes. Cytosolic Cerulean was expressed from a pEYFP-N1 plasmid where EYFP had been replaced with Cerulean.

For fluorescent imaging, HEK293A cells (ATCC, USA) were maintained in Dulbecco's Modified Eagle Medium (DMEM—Lonza, Switzerland) complemented with 10% fetal bovine serum (Biosera, France) and Penicillin/Streptomycin (100 U/ml and 100 µg/ml, respectively—Lonza, Switzerland). Cells were seeded on poly-L-lysine pretreated (0.001%, 1h) 24-well imaging plates (Eppendorf, Germany) at a density of 1e05 cells/well. On the next day, cells were co-transfected with plasmids coding cytoplasmic Cerulean and D4H-mVenus (0.25 µg/well each) using Lipofectamine 2000 (0.75 µl/well, Invitrogen, USA).

Image acquisition started 24h post-transfection, after the medium had been changed to 300 µl/well HEPES-buffered DMEM without phenol-red (Gibco, USA). Images were acquired automatically using the ImageXpress Micro Confocal High-Content Imaging System

(Molecular Devices, USA), with a 40x Plan Fluor objective. CFP-2432C and fluorescein iso-thiocyanate (FITC) filter sets were used for Cerulean and D4H-mVenus images, respectively, both with an exposure time of 300 ms. After acquiring control images (30 min), cells were treated with either DMSO (as control) or with the drugs indicated on Fig 4E in a volume of 100 μl/well (270 min). Measurements were performed at 30˚C. Three independent measurements were made, with duplicate wells for each condition and 5 images/well taken for each time point.

All chemicals used for treatment were purchased from Sigma-Aldrich Merck (Germany). Amiodarone HCl, chlorpromazine HCl, loperamid HCl and rosuvastatin calcium were dissolved in DMSO, stored at -20˚C as 10 mM stock solutions and diluted in cell medium promptly before cell treatment to a final concentration of 10 μM. MβCD was stored as powder at 4˚C and freshly dissolved in cell medium before treatment to a final concentration of 10 mM.

## Image analysis pipeline

Images were segmented with *Cellpose* Python library [45], which is a generalist, deep learning-based segmentation method. To select high-quality images the cytoplasm marker channel was used with Laplace filtering. We used high-quality images (filtered according to an appropriate upper threshold of standard deviation of Laplace value in each experiment) as input of the Cellpose model, with parameter channel set to greyscale and cell diameter greater than 200 pixels.

After identifying cell boundaries, we applied binary erosion (scipy Python library [72]) with default structure and 10 iterations to determine cytoplasm boundary, or binary dilation with default structure and 5 iterations to determine PM outer boundary. The boundary of PM was determined by subtracting the cytoplasm boundary from the outer boundary. We calculated the $\log_2$ ratio of the mean PM and mean intracellular D4H fluorescence intensities for each cell in the D4H channel to examine the changes of plasma membrane cholesterol distribution. For statistical analysis, we used log2(PM/IC) ~ Time + Time: Drug + Exp linear model, where Time corresponds to elapsed time after drug treatment, Drug factor represents the used drug, using DMSO as reference level. Exp factor represents the (n = 3) individual experiments.

## Viral infection and cholesterol rescue experiments

Amiodarone HCl (Sigma-Aldrich, Merck KGaA, Germany) was dissolved in DMSO (Sigma-Aldrich, Merck KGaA, Germany) and kept at -20˚C. Chlorpromazine (in house synthesized based on [73]) and loperamide HCl (Sigma-Aldrich, Merck KGaA, Germany) were freshly dissolved in water and filtered prior to the treatment. 10 mM stock solutions were made from the drugs. Vero-E6 cells were seeded in a 96-well plate on the day before the experiments. On the next day the cells were treated with 100 μl of 50 μM remdesivir or loperamide or 12 μM chlorpromazine or 6 μM amiodarone solution overnight. 1 hour prior to the infection the cell culture media containing the different drugs was replaced with media containing 80 μM cholesterol (Sigma-Aldrich, Merck KGaA, Germany). After the 1-hour-long cholesterol treatment the cells were infected with SARS-CoV-2 (GISAID accession ID: EPI_ISL_483637) at MOI:0.01 in a BSL-4 laboratory. Cells were incubated with the virus for 30 minutes then the media was replaced with fresh cell culture media. During the investigation (except cell seeding) DMEM (Lonza Group Ltd, Switzerland) supplemented with 1% Penicillin- Streptomycin (Lonza Group Ltd, Switzerland) and 2% heat-inactivated fetal bovine serum (Gibco, Thermo Fisher Scientific Inc., MA, USA) were used. 48 hours post infection (hpi) the cells were inspected under microscope and RNA was extracted from the supernatant (Zybio EXM 3000

Nucleic Acid Isolation System, Nucleic Acid Extraction Kit B200-32). Viral copy number was determined using droplet-digital PCR technology (Bio-Rad Laboratories Inc., CA, USA). SARS-CoV-2 RdRp gene specific primers and probe were utilized (Forward: GTGARATGGT CATGTGTGGCGG, reverse: CARATGTTAAASACACTATTAGCATA and the probe was: FAM-CAGGTGGAACCTCATCAGGAGATGC-BBQ). For statistical analysis, measured viral copy numbers were $\log_2$ transformed, and we used a log2(CV) ~ Drug $^*$ Cholesterol + Exp, where Drug factor represents the used drug (untreated as reference level), Cholesterol factor represents cholesterol replenishment treatment (no treatment as reference level). Exp factor corresponds to the (n = 4) individual experiments.

## Supporting information

**S1 Fig. Representative confocal microscopy images of D4H-mVenus transfected HEK293A cells treated with loperamide and rosuvastatin.**
(TIFF)

**S2 Fig. Determination of cytoplasm and membrane boundaries.** Cells are detected on the cytoplasm marker channel, then boundaries of cytoplasm and membrane are determined for each cell. The D4H channel is used for the calculation of the PM/IC ratio.
(TIFF)

**S3 Fig. Absence of marked toxic effects of used drugs in the tested concentrations.**
(TIFF)

**S1 Table. Drug signature similarities (Spearman's rho) to SARS-CoV-2 infection signatures (GSE147507 A549 SARS-CoV-2, GSE147507 Calu3 SARS-CoV-2 and GSE148729 Calu3 SARS-CoV-2 columns, respectively) and Random Forest Classifier based predicted probability of antiviral effect.**
(CSV)

## Acknowledgments

We thank Aurélien Dugourd, Gergő Gulyás, Kinga Kovács and András D. Tóth for the helpful discussions regarding the manuscript, and Péter Mátyus for his help in organising the collaborations between computational and experimental groups. We thank Kata Szabolcsi for technical assistance in the cholesterol sensor experiments, Katalin Gombos, Zsófia Lanszki and Balázs Somogyi for their help in the *in vitro* infection experiments, and Tamás Kálai for compound synthesis. On behalf of Project DRUGSENSPRED we thank for the usage of ELKH Cloud (https://science-cloud.hu/) that significantly helped us achieve the results published in this paper. Schematic figures were created with BioRender.com.

## Author Contributions

**Conceptualization:** Szilvia Barsi, Bence Szalai.

**Data curation:** Szilvia Barsi, Bence Szalai.

**Formal analysis:** Szilvia Barsi, Alberto Valdeolivas, Bence Szalai.

**Funding acquisition:** Péter Várnai, Ferenc Jakab, Bence Szalai.

**Investigation:** Szilvia Barsi, Henrietta Papp, Dániel J. Tóth, Anett Kuczmog, Mónika Madai, Péter Várnai, Bence Szalai.

**Methodology:** Szilvia Barsi, Alberto Valdeolivas, Bence Szalai.

**Project administration:** László Hunyady.

**Resources:** Dániel J. Tóth, Péter Várnai, Ferenc Jakab, Bence Szalai.

**Supervision:** László Hunyady, Péter Várnai, Julio Saez-Rodriguez, Ferenc Jakab, Bence Szalai.

**Validation:** Henrietta Papp, Dániel J. Tóth, Anett Kuczmog, Mónika Madai, Péter Várnai.

**Visualization:** Szilvia Barsi, Alberto Valdeolivas.

**Writing – original draft:** Szilvia Barsi, Bence Szalai.

**Writing – review & editing:** Szilvia Barsi, Henrietta Papp, Alberto Valdeolivas, Dániel J. Tóth, Anett Kuczmog, Mónika Madai, László Hunyady, Péter Várnai, Julio Saez-Rodriguez, Ferenc Jakab, Bence Szalai.

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
