## [Decision Letter · Decision Letter 0]

28 Dec 2021

Dear Dr. Szalai,

Thank you very much for submitting your manuscript "Computational drug repurposing against SARS-CoV-2 reveals plasma membrane cholesterol depletion as key factor of antiviral drug activity" for consideration at PLOS Computational Biology. As with all papers reviewed by the journal, your manuscript was reviewed by members of the editorial board and by several independent reviewers. The reviewers appreciated the attention to an important topic. Based on the reviews, we are likely to accept this manuscript for publication, providing that you modify the manuscript according to the review recommendations.

The reviewers were supportive of the overall study. There were minor concerns raised that should be addressed for clarification.

Sincerely,

James Costello

Guest Editor

PLOS Computational Biology

Feilim Mac Gabhann

Editor-in-Chief

PLOS Computational Biology

[LINK]

The reviewers were supportive of the overall study. There were minor concerns raised that should be addressed for clarification.

Reviewer's Responses to Questions

**Comments to the Authors:**

Reviewer #1: In this article, the authors introduced an interesting and counterintuitive technique for drug repurposing in for SARS-Cov-2: finding drugs that have signature similar to infection signature. The rationale is that infected cells activate adaptive, antiviral pathways, so that the relationship between effective drug and disease signature can be more ambiguous. In addition, as infected cells activate adaptive antiviral pathways, inhibiting these pathways does not decrease viral replication.

Based on the proposed analysis in this paper, three drugs (loperamide, amiodarone, and chlorpromazine) were identified as helpful in decreasing the plasma membrane cholesterol level, which in turn can have antiviral effects. The authors have provided sufficient analysis and experiments to support their claim. In addition, the manuscript is well-written. I only have three minor comments that would help to improve the quality of the article:

1. The authors used Random Forest Classifier to predict the effective drugs with the 100 trees. It would be great if the authors can provide justification why the choose RF over many other techniques? Also, why did they choose 100 as the number of trees? Would the conclusion be the same if they use other classification techniques or parameters?

2. I recommend that the authors briefly describe each section in the introduction. This might help readers know the purpose of each section so that they can refer directly to that section for a detailed explanation.

3. On page 9, the second paragraph refers readers to figure 2E. I believe that the authors meant 2D.

Reviewer #2: The authors use public data on changes in the gene expression of cell lines treated with antiviral drug candidates and cell lines infected by SARS-COV-2, showing that similarities in the shifts of expression levels can be predictive of the effectiveness of drugs. For some drug candidate they then test a causal mechanism identified by the similarity-based analysis and find that the antivirals reduce cholesterol content of the plasma membrane and their effect can be reverted by cholesterol replenishment.

I find the second experimental part is a nice confirmation of the computational methods of the first part, and it is in this 2nd part that the main novelty of the paper lies. The computational methods used in the article were previously developed, so I don't see a methodological innovation there, except on the conceptual level that the authors used similarity and not dissimilarity of GE changes between drug-treated and infected cells to make drug effect predictions. In this sense i'm not sure PLoS Comp Biol is the best venue for this paper, but this is for the editors to decide. Since using public data anyhow in the 1st part of the article I think the authors could check their similarity based prediction for other viruses if data is available, ie. can drug effects be predicted for other viruses too based on similarity? Overall I found the article interesting and convincing for the specific hypothesis made re SARS-CoV-2, to make a general argument for similarity-based predictions though I find the sample size might be too small at this point.

Some comments on figures and the text:

p6: AKT1 and MAPK1 are among the most connected nodes in PPI/gene regulation networks. Is this specific to sars-cov-2 at all? Comparison with other viruses could be useful here.

On p11-12 the authors say the prediction performance based on gene expression (GE) similarities between infection and antivirals is 'reasonable'. It would be good to have a comparison with other studies where such predictions were validate, ideally not just in vitro. Are these ROC AUC values typical for such studies? Or higher?

Fig3B: since there seems to be data on gene expression shifts for other viruses, could the similarity based prediction be applied to some of these? For instance can effectiveness of flu drugs be predicted from the similarity of GE shifts in drug-treated and infected cells? If so, this could be a further argument for the method.

In Fig3/C what do the diamonds correspond to? (and also the intervals - CI50 and CI95? How were these intervals calculated?)

In Fig4/C it's not clear to me what is meant by predictive probability exactly and why the highest values here are 0.1-0.2. This is a probability with respect to which model? How do these values compare to other drug effect prediction studies?

In Fig 4/D i can see well that there are a lot more intracellular cholesterol on the panels with the (predicted) effective drugs. However that there's less cholest. in the cell plasma membrane is less clear by the eye at least. We see too few cells in any case, so I was not convinced by this picture. I think the next panel makes the point much better.

p14 and Fig4E: Just by eye the amiodarone curve looks very close and within the range of variation of the control. Somewhat surprised by the extremely low p-value. Is the p-value for the coefficient of the linear fit rather than on whether there's a difference wrt the control? Can the authors explain this? Maybe a significance test on the difference with DMSO would be better, as this one also shows a downward trend (by eye).

Fig5B: is the bar chart on the right an illustration or actual data? if the latter there should be error bars. If it's an illustration I am not sure it is needed.

Minor comments:

0: remove paperpile links in file

p4: COVID-19 disease burden. These are the *reported* numbers; real # of deaths likely to be higher. Refer to excess mortality estimates or say 'reported deaths'.

p9: refers to Fig 2E, which doesn't exist. I assume it's Fig2D?

p15: I don't find Fig4A intuitive, specifically that the drugs (might) cause cholesterol depletion. I think this should be more explicitly shown somehow (eg. with arrows)

Reviewer #3: Summary

The authors make a complete and through exploration and analysis to obtain the activity of pathways and TF in covid infected cells. Their pipeline investigates the pathways and TFs activities of different cells and compare with the activity of drug induced cells, is very clear and straightforward.

They revealed that the similarity of the TF activities in infected cells and drug induced cells are an important feature to determine the efficiency of the drug treatments for SARS-coV-2 in vitro (although not true for all effective drugs). They as well demonstrated quantitatively that the similarity to infection signatures is good to predict effective drugs. This result is important since it can be explored in the search for treatments of another virus.

The main result from the comparison of the T.F. activities, infected vs drug treated cells, is that regardless of their similarity, for some clusters of drugs, they observe a high activation of key regulators of cholesterol, SREBFs. Which is surprising given that high cholesterol synthesis can stimulate viral entry.They hypothesize a mechanistic scenario for this contra intuitive result: The anti-SARS-CoV-2 drugs decrease plasma membrane cholesterol, which in turn activate SREBFs TFs.

To prove this, they performed a series of experiments to demonstrate the role of the membrane cholesterol in the antiviral effect of 3 different drugs (efficient drugs against SARS-coV-2).

Comments and questions:

-I did not have clear what was the input of the Random Forest Classification, since one result obtained from it is the importance of HNf4A, TFAP2A and TP63 transcription factors, but they are not presented in Fig.1B nor Fig.2B as to have available their activities under drug induction. The authors could explain why the lack of such data.

--They were able to demonstrate the role of membrane cholesterol importance for the drug antiviral effect in only 1 out of 3 investigated drugs. Even so, they show (Fig.4C) that the depletion of membrane cholesterol is sensitive to the elapsed time after drug induction, therefore time can be a key parameter, specially in pre-treatment of cells, for the cholesterol localization and its effect once the cells are infected. Therefore, this encourages to a further study of the mechanism of the antiviral effect of drugs over time and the relationship with cholesterol.

--Do the authors have an explanation on why Chlorpromazine shows similar viral copies as in the control (Fig. 5C), with and without cholesterol replenishment?

Minor note:

-Numbering in Fig.4, the numbering A and B are cut and the x-label od 4.C is missing (is the elapsed time in minutes)?

I appreciate the complete work done in the paper: it includes a curation of previous experimental data, network analysis, statistical analysis, and finally, they performed well focused experiments to try to demonstrate their hypothesis.

Overall, I think the article presents a good study and demonstrates the important role of plasma membrane cholesterol in the antiviral effect of some drugs against SARS-Cov-2. It also gives good information (and technique) to evaluate the role of other factors and mechanism for drug and for infection, as they present the activities of many other pathways and transcription factors. This can be helpful not only in the focus of the role of cholesterol but be complementary of other studies, as the mechanisms of other repurposing drugs.

Hence, with the above major comment and a few minor comments addressed I found that this manuscript deserves be published in PLOS Computational Biology.

Reviewer #4: Barsi et al performed computational analyses on transcriptomics data from in vitro SARS-CoV-2 infected cell lines and from cell lines treated with drugs showing anti-SARS-CoV-2 activity. Using functional genomics study and machine learning, the authors identified SREBF1 and SREBF2 transcription factors that regulate lipid metabolism are activated by effective drugs. The authors hypothesized and further showed drug-induced decreased level of plasma membrane cholesterol via depleted cholesterol metabolism has anti-viral effect. Their experimental validations showed amiodarone, a drug that decreases plasma-membrane cholesterol content is a potential repurposing drug candidate to combat SARS-CoV-2.

This is an interesting study and the work connects depleted plasma membrane cholesterol induced by drugs exert anti-SARS-CoV-2 activities and open a new door to repurpose existing drugs to combat COVID-19. I have no major concern on the results on this work. The work is deemed publishable with few minor concerns being addressed:

(1) There is a number of reported works on the role of cholesterol (including SREBF) with COVID-19, the authors should mention and discuss some of them to provide a clearer picture how the present work fills the gap in our understanding and treatment of COVID-19.

(2) Although the authors showed depleted plasma membrane cholesterol is essential for anti-viral activities and described increased SREBF1/2 as a compensatory consequence, the authors might need to provide further elaboration on the following points: (i) Is inhibition of SREBF1/2 an adaptive response of infected cells to restrict the supply of cholesterol to prevent viral entry (i.e., a protective mechanism of infected cells against virus)? (ii) Is increased SREBF1/2 by effective drugs a homeostatic (or protective) response of cell to counter cytotoxic effect of effective drugs that lead to decrease plasma membrane cholesterol?

(3) The usage of the term “drug signature” is confusing as it can be understood as drug-induced gene expression patterns or chemical signature of drug. The authors need to define this term more explicitly in the introduction.

(4) Revise subtitle of section 2.5 to “Supplementing cholesterol reverse anti-SARS-CoV-2 activity of amiodarone”.

**Have the authors made all data and (if applicable) computational code underlying the findings in their manuscript fully available?**

Reviewer #1: Yes

Reviewer #2: Yes

Reviewer #3: None

Reviewer #4: Yes

PLOS authors have the option to publish the peer review history of their article (what does this mean?). If published, this will include your full peer review and any attached files.

Reviewer #1: No

Reviewer #2: No

Reviewer #3: No

Reviewer #4: No

Figure Files:

Data Requirements:

Reproducibility:

References:

---

## [Decision Letter · Decision Letter 1]

15 Mar 2022

Dear Dr. Szalai,

We are pleased to inform you that your manuscript 'Computational drug repurposing against SARS-CoV-2 reveals plasma membrane cholesterol depletion as key factor of antiviral drug activity' has been provisionally accepted for publication in PLOS Computational Biology.

Best regards,

James Costello

Guest Editor

PLOS Computational Biology

Feilim Mac Gabhann

Editor-in-Chief

PLOS Computational Biology

All reviewers are satisfied with the revision.

Reviewer's Responses to Questions

**Comments to the Authors:**

Reviewer #1: The authors have satisfactorily addressed my comments.

Reviewer #3: I thank for the ansewers to my questions and the clarification for different points. I see the different mistakes/typos noted from me and other reviewers were ammend.

Reviewer #4: I have no more questions on revised paper.

**Have the authors made all data and (if applicable) computational code underlying the findings in their manuscript fully available?**

Reviewer #1: Yes

Reviewer #3: None

Reviewer #4: Yes

PLOS authors have the option to publish the peer review history of their article (what does this mean?). If published, this will include your full peer review and any attached files.

Reviewer #1: No

Reviewer #3: No

Reviewer #4: No

---

## [Editor Report · Acceptance letter]

6 Apr 2022

PCOMPBIOL-D-21-02092R1 

Computational drug repurposing against SARS-CoV-2 reveals plasma membrane cholesterol depletion as key factor of antiviral drug activity

Dear Dr Szalai,

I am pleased to inform you that your manuscript has been formally accepted for publication in PLOS Computational Biology. Your manuscript is now with our production department and you will be notified of the publication date in due course.

With kind regards,

Katalin Szabo
